# Analysis of In Vivo Transcriptome of Intracellular Bacterial Pathogen *Salmonella enterica serovar Typhmurium* Isolated from Mouse Spleen

**DOI:** 10.3390/pathogens10070823

**Published:** 2021-06-30

**Authors:** Na Sun, Yanying Song, Cong Liu, Mengda Liu, Lanping Yu, Fangkun Wang

**Affiliations:** 1Department of Preventive Veterinary Medicine, College of Veterinary Medicine, Shandong Agricultural University, 61 Daizong Street, Taian 271018, China; sunnasdau@163.com (N.S.); songsdau@163.com (Y.S.); liucongsdau@163.com (C.L.); 2Shandong Provincial Key Laboratory of Animal Biotechnology and Disease Control and Prevention, Shandong Agricultural University, 61 Daizong Street, Taian 271018, China; bolilala_715@126.com; 3Laboratory of Zoonoses, China Animal Health and Epidemiology Center, 369 Nanjing Road, Qingdao 266032, China; liumengda@cahec.cn

**Keywords:** *Salmonella* *enterica serovar Typhimurium*, RNA-seq, comparative transcriptomics, pathogenic mechanism

## Abstract

*Salmonella* *enterica serovar Typhimurium* (*S. Typhimurium*) is an important intracellular pathogen that poses a health threat to humans. This study tries to clarify the mechanism of *Salmonella* survival and reproduction in the host. In this study, high-throughput sequencing analysis was performed on RNA extracted from the strains isolated from infected mouse spleens and an *S. Typhimurium* reference strain (ATCC 14028) based on the BGISEQ-500 platform. A total of 1340 significant differentially expressed genes (DEGs) were screened. Functional annotation revealed DEGs associated with regulation, metabolism, transport and binding, pathogenesis, and motility. Through data mining and literature retrieval, 26 of the 58 upregulated DEGs (FPKM > 10) were not reported to be related to the adaptation to intracellular survival and were classified as candidate key genes (CKGs) for survival and proliferation in vivo. Our data contribute to our understanding of the mechanisms used by *Salmonella* to regulate virulence gene expression whilst replicating inside mammalian cells.

## 1. Introduction

*Salmonella* is a Gram-negative pathogen that poses a health threat to humans and livestock. According to the World Health Organization (WHO), salmonellosis is the most common food-borne disease, with nearly one-tenth of the world’s population becoming infected each year, with around 33 million deaths (http://www.who.int/mediacentre/factsheets/fs139/en/, accessed on 20 February 2018). Among the more than 2600 documented serotypes, *Salmonella enterica serovar Typhimurium* is the predominant serotype associated with salmonellosis worldwide [1,2].

Research has largely focused on *S. Typhimurium*, which is a natural pathogen of mice, to understand the pathogenesis of typhoid fever in a mouse model of systemic salmonellosis [3]. The mechanism of survival and reproduction in vivo may be key to studying the pathogenesis of *S. Typhimurium*. During the infection process, *S. Typhimurium* must adjust and adapt to rapidly changing intracellular and extracellular environments by downregulating the expression of non-essential genes, upregulating the expression of the genes necessary for survival, and expressing essential virulence genes [4,5,6,7]. Many of these adaptive processes are regulated at the transcriptional level [8]. In recent years, many studies have confirmed that many bacteria that induce the high expression of genes in vivo are pathogenic [9,10,11,12,13]. Type III secretion systems (T3SSs) are central to the interactions of many pathogenic bacteria with their hosts [14]. The *Salmonella* pathogenicity-island (SPI)-1- and -2 encode T3SSs, which are important for the invasion of host cells and intracellular replication [15]. Studies have shown that some important invasion-related T3SS proteins, effector proteins such as AvrA, SipB, and SipC, are only detected during bacterial infection [16,17]. In studies investigating the pathogenesis of *Salmonella*, the induction of highly expressed genes in vivo is a research focus [18,19,20,21].

In this study, we systematically screen and identify significant differentially expressed genes of intracellular bacteria in host spleens [22]. We study the bacterial transcriptome rather than the proteome to avoid difficulties in separating host cell proteins and apply a new generation of high Flux BGISEQ-500 expression profiling [23] to obtain *S. Typhimurium* transcriptome data for intracellular and extracellular cultures. RNA sequencing (RNA-seq) analysis showed that the transcription levels of genes were significantly altered in the pathogenic bacteria in vivo. We screen and identify key factors in the intracellular proliferation of *S. Typhimurium* by comparative transcriptome analysis combined with literature mining. Our results will help to reveal novel mechanisms employed by *S. Typhimurium* to adapt and survive in vivo and yield novel candidate key genes (CKGs) for the survival and proliferation of this pathogen in vivo.

## 2. Results

### 2.1. Gene Expression Profiles

Three samples of *S. Typhimurium,* taken from the spleens of infected mice and three in vitro control groups, were set up, respectively. A total of six samples were analyzed by high-throughput sequencing using the BGISEQ-500 platform, with an average yield of 21.95 M of data per sample. TN6_2A, TN6_3A, and TN6_4A were the three in vivo experimental replicates, and TW6_2A, TW6_3A, and TW6_4A were three in vitro controls. The quality of the filtered reads is shown in the filtered reads quality statistics table (Appendix A). After obtaining the clean reads, we used HISAT to compare the clean reads to the reference genome sequence. A total of 4479 genes were detected. Venn diagrams were generated to quantify the overlap between the three biological replicates. Among them, 4477 transcripts were detected in the in vitro group, and 4460 genes were identified among the three biological replicates (Appendix A). In total, 3737 transcripts were detected in the in vivo group, of which 2853 genes were identified among the three biological replicates (Appendix A). To determine the correlation in gene expression between the samples, the Pearson correlation coefficient of all gene expression levels between each sample was calculated. The correlation coefficient between in vivo samples was more than 84.2%, compared with more than 93.3% for the in vitro samples (Appendix A). To show the number of genes in each FPKM group (FPKM ≤ 1, FPKM 1–10, FPKM ≥ 10) for each sample, we performed statistical analysis; the results are shown in Appendix A.

### 2.2. Identification of DEGs

To determine the DEGs among the 4479 genes detected by RNA-seq, DEGseq software was used, with the difference multiple set to more than twice and a Q-value ≤0.001 set as the screening parameter. Comparative transcriptomic analysis (in vivo vs. in vitro) revealed a total of 1340 DEGs (Appendix A), of which 633 were upregulated in expression and 707 were downregulated in expression (Figure 1).

### 2.3. Gene Ontology (GO) Enrichment Analysis for DEGs

Gene ontology divided the genes into three major functional categories: molecular function, cellular component, and biological process (Figure 2a). The molecular function classification results for the DEGs showed that they were mainly distributed into the following categories: binding, catalytic activity, transporter activity, and transcription regulator activity. The cellular component classification results for the DEGs showed that they were mainly distributed into the following categories: membrane, membrane part, protein-containing complex, and extracellular region. The biological process classification results for the DEGs showed that they were mainly distributed into the following categories: cellular process, metabolic process, localization, and response to stimulus.

The enriched bubble map shows the enrichment of GO terms from three dimensions. The figure below shows the GO enrichment results for the DEGs (Figure 2b). GO functional enrichment analysis of the DEGs revealed that the DEGs were significantly enriched in the following categories: pathogenesis, interspecies interaction between organisms, multi-organism process, locomotion, and extracellular region (Appendix A).

### 2.4. Pathway Enrichment Analysis for DEGs

The gene-involved KEGG metabolic pathway is divided into seven branches: Cellular Processes, Environmental Information Processing, Genetic Information Processing, Human Disease, Metabolism, Organismal Systems, and Drug Development. Further classification statistics are performed under each branch (Appendix A). The KEGG pathway annotation classification results for the DEGs are shown in Figure 3a.

KEGG pathway enrichment analysis showed that the DEGs were significantly enriched in multiple pathways (Appendix A), such as microbial metabolism in diverse environments, bacterial chemotaxis, *Salmonella* infection, bacterial secretion system, two-component system, and flagellar assembly (Figure 3b). Almost all genes enriched upon bacterial chemotaxis and flagellar assembly were significantly downregulated, while those related to the biosynthesis of siderophore group nonribosomal peptides were significantly upregulated (Appendix A). For instance, *Salmonella* expresses flagellin when replicating in vitro but reduces the expression of this protein during intracellular growth in a mammalian host [24,25]. Our results closely match these published results. Intracellular *S. Typhimurium* repressed the transcription of fliC and also downregulated its positive regulator, fliA.

### 2.5. Validation of the RNA-seq Data by qPCR

In order to verify the reliability of RNA-seq through literature mining and data analysis, 16 genes were selected from genes with significant upregulation and downregulation (Figure 4a). Using the *16S* gene (Figure 4b) and the *pdxJ* gene (Figure 4c) as internal controls, RT-qPCR analysis was performed to validate the expression profiles of DEGs in *S. Typhimurium*-infected mice. Fifteen genes, except for *menG*, were significantly upregulated or downregulated, consistent with the RNA-seq data. The overall coincidence rate was 93.75%.

### 2.6. Screening of CKGs for Survival and Proliferation In Vivo

A further in-depth analysis of the DEGs was conducted, and 58 genes that were more than 10 times upregulated in vivo (FPKM > 10) were excluded. These proteins are mainly T3SS proteins, pathogenicity-island-2-secreted effector proteins, metabolic enzymes, regulatory factors, transport proteins, and hypothetical proteins. Among them, the T3SS-associated protein had the largest number of proteins, accounting for 29.3% (17/58).

Through a literature review, we eliminated the genes reported to be associated with intracellular survival and proliferation in *Salmonella*. Finally, 26 of the 58 genes were classified as CKGs for survival and proliferation in vivo (Table 1).

## 3. Discussion

The infection transcriptome profile of *S. Typhimurium* in vivo is significantly different from that under laboratory growth conditions. In this study, a total of 1340 DEGs were screened through quantitative gene analysis and subjected to various analyses based on gene expression levels (principal component, correlation, differential gene screening). These DEGs were further analyzed by GO function significant enrichment analysis, pathway significant enrichment analysis, and clustering analysis, and most of the genes found to be enriched were classified into cellular processes and catalytic activity, structural constituents of ribosome, structural molecule activity, pathogenesis, locomotion, bacterial chemotaxis, flagellar assembly, and *Salmonella* infection. These DEGs, therefore, appeared to be functionally closely related to the adaptation of *Salmonella* to in vivo microenvironments, ensuring the survival of the bacteria and promoting bacterial proliferation. On this basis, we optimized the screening conditions and screened 58 significantly upregulated genes (FPKM > 10). These DEGs mostly encoded T3SS proteins, some enzymes, and the SPI2 effector proteins. T3SS effectors provide Gram-negative bacteria with a unique virulence mechanism, enabling them to inject bacterial effector proteins directly into the host cell cytoplasm, bypassing the extracellular milieu [58]. *Salmonella* replicates within both non-phagocytic epithelial cells and macrophages in *Salmonella*-containing vacuoles (SCVs) [59]. T3SS secretion effectors alter vacuole positioning by acting on host cell actin filaments, microtubule motors, and components of the Golgi complex. Once positioned, maturation is stalled, and bacterial replication is initiated. The T3SS effectors can block phagocytosis or promote bacterial invasion of non-phagocytic cells, altering membrane trafficking and modulating innate and adaptive immune responses [60,61]. The natural immune system is the first line of defense against bacterial disease [62]. These effectors, which are injected into host cells through a secretory system (such as T3SS), employ complex and sophisticated strategies to block and control the host’s signal transduction pathways, particularly those that have important functions in host innate immunity. For example, *Salmonella* AvrA can be secreted into the host cytoplasm, where it inhibits inflammation and cellular pro-apoptotic responses [17,63]. SseL protein can inhibit the intracellular NF-κB pathway and enhance the pathogenicity of *Salmonella pullorum* [21]. These activities enable bacteria to survive and replicate, causing disease.

In this study, most of the proteins in the T3SS are upregulated in vivo, and their secreted effector proteins, such as SpiC, SseB, SseC, SseD, SseE, SseF, SseL, and STM1410, also showed significant high levels of expression. In the Sse family, in addition to SseA, which is essential for bacterial survival in cells and a key factor in determining bacterial virulence [34], SseJ also regulates SCV membrane dynamics and is associated with SCV and Sifs [55]; SseB, SseC, and SseD are located on the bacterial cell membrane and are also essential for the survival of bacteria in cells, being secreted by the SPI2-mediated T3SS and playing a toxic role outside the bacterial membrane [64]. SsaM is essential for the secretion of SseB, SseC, and SseD. Following the deletion of SsaM, the bacteria cannot form Sifs and cannot transfer the effector protein SseJ to infected cells. The virulence of the deletion strain and its replication ability in infected cells are all decreased [28]; Spic also promotes SseB and SseC secretion. The decrease in virulence caused by Spic deletion is not due to a single effector deletion but results from the deletion of all SPI2 effector proteins. Spic inhibits the interaction between SCVs and late endosomes and lysosomes, as well as the endocytosis and recycling of transferrin [38], so *SsaM* and *Spic* are also virulence genes of *S. Typhimurium* [37].

To ensure intracellular survival, *Salmonella* must not only avoid removal by the host immune system but also strive to obtain nutrients, which requires the adjustment of gene expression levels to adapt to a rapidly changing environment [7]; these processes are mostly dependent on certain special biological macromolecules. At present, it is still unclear which regulatory networks are affected by effector proteins, the SPI, and transport processes and what their roles are in adapting to the SCV microenvironment in the host cell [7].

Biotin is an essential enzyme cofactor that is necessary for essential steps of central metabolism, including fatty acid synthesis and amino acid degradation [65]. To date, the biotin synthesis pathway has been studied in detail in *E. coli*, *Bacillus sphaericus*, *Bacillus subtilis*, *Saccharomyces cerevisiae*, and plants [66]. Fis is a key DNA-binding protein that plays a central role in coordinating the expression of metabolic factors, including biotin synthesis, and T3SS secretion factors [67]. The genes involved in biotin synthesis (bioB, bioC, and bioF) were highly upregulated in the *S. typhimurium* fis mutant, which was grown in LB broth. In our study, both fis- and biotin-synthesis-related genes were upregulated in vivo. Studies have shown that after *S. Typhimurium* infection of macrophages, SPI2-mediated secretion of the T3SS proteins SsaQ and SseE and expression of the biotin biosynthesis proteins BioB and BioD increased. After the loss of the bioB gene, the ability of the bacteria to proliferate in phagocytes was reduced, indicating that biotin plays a role in the survival of *S. Typhimurium* in macrophages [43]. In our experiments, not only bioB and bioD were found to be upregulated but also bioA, bioC, and bioF. BioF, one of the CKGs, is an enzyme that catalyzes the first step in the biotin synthesis pathway. Its structure and function have been well studied in microorganisms such as *E. coli*, *B. subtilis*, and *B. sphaericus* [68,69], but not in *S. Typhimurium* [43].

In host SCVs [70], a variety of metabolic changes, including bacterial oxidation and nitrification, are initiated to accommodate changes in the microenvironment [71]. Physiological, metabolic, and effector protein-mediated adaptation strategies allow the bacteria to replicate within the SCVs and to form persister cells [70,72]. Sulfur is an essential element for microorganisms that can be obtained from a variety of compounds, with sulfate being the preferred source [73]. Sulfate uptake is carried out by sulfate permease belonging to the SulT (CysPTWA), SulP, CysP / (PiT), and CysZ families. The main proteins of the sulfur metabolism transport pathway are significantly upregulated in cells, which may be an adaptation to the SCV microenvironment. Among them, CysA is a sulfate/thiosulfate transfer ATP-binding protein that is involved in the formation of the ABC transporter complex CysA WTP (located inside the cell membrane) and is responsible for energy coupling with the transport system. Within the bacterial cytoplasm, CysH is a phosphate adenosine phosphate reductase (thioredoxin) that is a member of the PAPS reductase family. The three substrates of the CysH enzyme are adenosine 3′,5′-diphosphate, sulfite, and thioredoxin disulfide, and the two products are 3′-adenosyl sulfate and thioredoxin. In a study of the intracellular pathogenic bacteria *M. tuberculosis*, it was confirmed that *cysA* deficiency leads to sulfate auxotrophy [74]. The activity of *cysH* can protect bacteria against free radicals during chronic infection, and nitrosation and oxidation exert excitatory resistance, which may be the mechanism that guarantees chronic persistent infection [75]. Nitrosation and oxidative stress play key regulatory roles in inflammation and the immune response. Nitric oxide produced by the host’s immune cell NO synthases has a severe impact on the major carbon metabolic pathways of *Salmonella* [76]. Following *Salmonella* infection in mice, both *cysA* and *cysH* were upregulated, suggesting that *Salmonella* has a similar mechanism of action to *M. tuberculosis* and plays an important role in regulating host cell nitrosation and oxidative stress.

Through GO analysis for DEGs, the molecular function classification results showed that there were only 52 significant DEGs distributed into transcription regulator activity (Appendix A). The levels of 20 transcription factors were upregulated, and 32 transcription factors were downregulated in vivo. For example, STM0859 was one such upregulated transcription factor, and hilC, hilD, and sprB were three downregulated SPI1 regulators in vivo. During infection, transcription factors STM0859 and leuO were co-regulated with the SPI-2 T3SS [77]. The transcriptional start sites of the STM0859 transcripts were only expressed in host macrophages and not in any in vitro conditions [78]. Srikumar et al. observed that the downregulation of SPI1 regulators hilC, hilD, and sprB correlated with the expression of their respective regulons. Our results closely match these published results.

Through our literature review of these significantly upregulated genes, we obtained a total of 26 CKGs that have not yet been reported to be related to intracellular survival and proliferation (Table 1). These genes are highly likely to be associated with such functions and are candidate key proteins for intracellular survival and proliferation. In future studies, we will further elucidate the expression of these proteins in host cells and their possible interaction with host cell proteins, enabling us to explore the immune signaling pathways that are involved in vivo. This will help us to clarify key issues such as the pathogenesis of intracellular bacteria and the mechanisms of evading host immune responses. Furthermore, such studies may provide target proteins for vaccine development and drug development, which may aid efforts to control infections by *Salmonella* and other important intracellular pathogens.

## 4. Materials and Methods

### 4.1. Bacterial Strains, Plasmids, and Animals

*S. Typhimurium* strain ATCC 14028, used in this study, was obtained from the American Type Culture Collection (ATCC; Manassas, VA, USA) and subcultured weekly on Luria–Bertani (LB) broth agar (LBA; Hopebio, Qingdao, China). Strains were maintained in cryovials at −80 °C. Colony-forming units (CFUs) were determined by plating on LB broth agar and incubating at 37 °C for 12 h.

Female BALB/c mice of 4 to 6 weeks of age were purchased from the Experimental Animal Center of Shandong University. All mice were housed in cages (5 mice/cage, 225 mm × 340 mm × 155 mm) with wood chips under specific-pathogen-free conditions in the animal facility, without psychological trauma or unnecessary pain throughout the study. All mice were euthanized with an overdose of isoflurane (5%). Exposure to isoflurane (5%) was continued for at least 1 min after breathing had ceased.

### 4.2. BALB/c Mice Infection and RNA Extraction

*S. Typhimurium* strain ATCC 14028 was grown overnight at 37 °C in LB broth, harvested by centrifugation at 1500× *g* for 10 min, and then washed twice in ice-cold (4 °C) phosphate-buffered solution (PBS; pH 7.2) and diluted with PBS. Each mouse in the experimental group was intraperitoneally injected with 2 × 10^4^ CFU of *S. Typhimurium*. At the same time as the intraperitoneal injection of the mice, 1 mL of the same cultured reference strain of *S. Typhimurium*, under laboratory growth conditions, was taken and centrifugated at 12,000 rpm at 4 °C for 15 min. The supernatant (in vitro sample) was discarded, and the reference sample was deposited at −80 °C.

Referring to Mahan’s method, three days after intraperitoneal (i.p.) injection, in vivo *Salmonella* were isolated from 10 female BALB /c mice spleens as one in vivo sample [22]. The bacteria in the in vivo group were separated from the mouse spleens entirely on ice. According to the literature [79], the infected spleens were homogenized with a Dounce glass ice bath precooled homogenizer, and the spleen homogenate was filtered through a 400-mesh cell strainer to remove tissue debris. The homogenate was added to an equal volume of 0.2% Triton X-100 solution and placed on ice. The captured bacteria were pelleted by centrifugation, and the contaminating murine RNA (due to cell lysis) was removed. Once the supernatant containing contaminating RNA was removed, the bacterial pellet was resuspended in RNA to stabilize the bacterial cells until RNA snap isolation.

Total RNA was isolated from the sample using the MiniBEST Universal RNA Extraction Kit (TaKaRa, Dalian, China). RNA purity and integrity were measured by the absorbance ratios (A260/280 and A260/230) and 2% agarose gel electrophoresis, respectively. In addition, the RNA integrity number (RIN) was determined using the Agilent 2100 Bioanalyzer (Agilent Technologies, Palo Alto, CA, USA) [80]. Three experimental repetitions were performed for each RNA extraction. Total RNA was stored at −80 °C prior to cDNA library construction.

### 4.3. Library Construction and RNA Sequencing

The total RNA was digested to remove any DNA contamination, and we removed contaminated mice RNA using OligodT magnetic beads to enrich the mRNA with polyA tails. rRNA reagent was then used to remove rRNA from total RNA. Interrupting reagent was then added to the sample, and an appropriate temperature and reaction time were applied to fragment the mRNA. The interrupted mRNA was then added to a one-strand synthesis reaction system, and one strand of cDNA was synthesized using a PCR cycler. Double-stranded cDNA was then synthesized using a two-strand synthesis reaction system under the appropriate temperature and reaction time to ensure the repair of the ends (addition of A bases at the 3′ ends) of the double-stranded cDNA. PCR was performed to amplify the ligands. An Agilent 2100 Bioanalyzer (Agilent DNA 1000 Reagents) was used to detect the range of inserted fragments in the library, and the ABI StepOnePlus real-time PCR system (TaqMan Probe) was used to quantify the concentration of the library. Using Illumina’s sequencing technology, the library was denatured into single chains by adding NaOH, then diluted appropriately. The denatured and diluted library was added into the FlowCell, hybridized with the junction on the FlowCell, and then, bridged PCR amplification was performed on the cBot-generated platform. Finally, the FlowCell was sequenced using the Illumina sequencing system and the SBS reagent.

### 4.4. Identification of Differentially Expressed Genes (DEGs)

The filtering software SOAPnuke, independently developed by BGI (Shenzhen, China), was employed for statistical analysis. The reads containing the linker, those with unknown base N contents greater than 5%, and low-quality reads were removed. The filtered “clean reads” were saved in the FASTQ format.

The clean reads were aligned and mapped to the *S. Typhimurium* reference genome (Assembly: GCF_000006945.2 _ASM694v2) using HISAT2 (version 2.1.0) under the spliced mapping algorithm with default parameters [81]. Bowtie2 [82] was used to compare clean reads to the reference gene sequences, and then RSEM [83] was used to calculate gene and transcript expression levels. The expression abundance of the mapped gene reads was normalized by fragments per kilobase of exon model per million mapped reads (FPKM) using StringTie (version: 1.3.0) with default parameters [84,85]. The mean FPKM value of three replicates in each group was calculated. For DEG analysis, three pairwise comparisons (in vivo vs. in vitro) were conducted. The DEGseq method was based on Poisson distribution, and DEG detection was performed according to the method described in the literature [86]. Using the statistical model described above, DEGseq proposed a new MA-plot-based application to determine the accuracy of DEGs. We defined genes with a fold difference of more than two and a Q-value ≤ 0.001 as DEGs and screened for significant DEGs.

### 4.5. Gene Ontology (GO) and Pathway Enrichment Analysis for DEGs

The Gene Ontology initiative (GO) describes the functional framework and conceptual set of genes of all biological products. It is specifically designed to support computational representations of biological systems [87]. KEGG is a database that includes genes and genomes. Over the years, KEGG has become a comprehensive knowledge base that assists in the biological interpretation of large-scale molecular datasets [88]. The DEGs (in vivo vs. in vitro) were subjected to functional classification using GO and KEGG analysis, respectively. The GO and KEGG pathways were considered significantly enriched when the *p*-value was <0.05. The *p*-value was then calculated from the normal distribution.

### 4.6. Quantitative PCR Analysis

Using the extracted *S. Typhimurium* RNA as a template, cDNA strands were synthesized under the action of reverse transcriptase according to the PrimeScript ™ RT reagent kit with gDNA Eraser (Perfect Real Time) from TaKaRa, followed by reverse transcription. The reaction product was stored at −80 °C. We randomly selected 16 DEGs for fluorescence quantitative PCR verification. Two pairs of primers, *16S* and pyridoxol 5′-phosphate synthase (*pdxJ*) [89], were used as internal references. Three biological replicates were performed for each gene, and the data were normalized to the internal reference gene and calculated using the 2 ^−ΔΔCT^ method [90]. The primers used for real-time qPCR analysis are described in Appendix A.

### 4.7. Statistical Analysis

For the analysis of differential genes, differential gene functional classification and pathway enrichment analysis were all enriched using the phyper function in R software. The formula for *p*-value calculation was divided into:(1)$$p=1-\overset{m-1}{\underset{i=0}{\sum }}\frac{\left(\begin{array}{c} M\\ i \end{array}\right)\left(\begin{array}{c} N-M\\ n-i \end{array}\right)}{\left(\begin{array}{c} N\\ n \end{array}\right)}$$

The *p*-value was corrected by FDR, and when FDR ≤ 0.01, this was regarded as significant enrichment.

In the knockout of candidate genes, data analysis was performed using *t*-test analysis.

## Figures and Tables

**Figure 1 pathogens-10-00823-f001:**
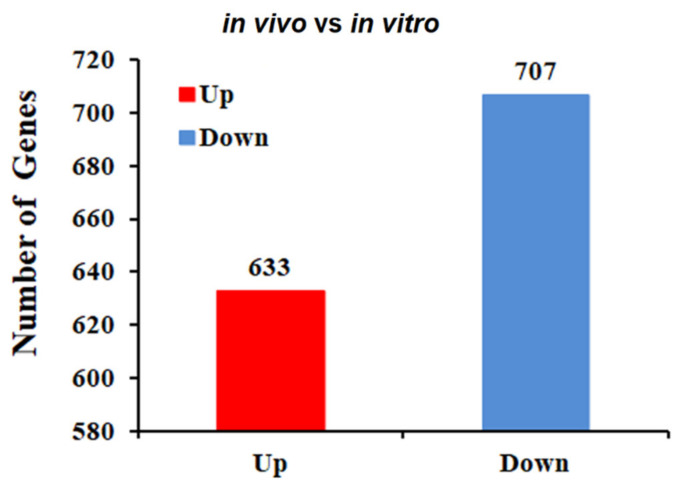
Differentially expressed genes between the in vivo and in vitro groups. Comparative transcriptomic analysis (in vivo/in vitro) showed a total of 1340 DEGs, of which 633 were upregulated in expression and 707 were downregulated in expression.

**Figure 2 pathogens-10-00823-f002:**
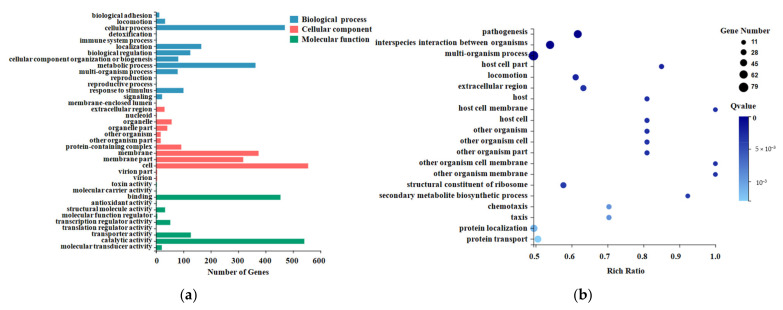
GO classification and enrichment analysis of the DEGs. (**a**) The X-axis represents the number of genes annotated for the GO entry, and the Y-axis represents the GO functional classification. (**b**) The X-axis represents the enrichment ratio, and the Y-axis represents the GO term. The size of the bubble indicates the number of differential genes annotated to a certain GO term. The color represents the enriched Q-value. The darker the color, the smaller the Q-value.

**Figure 3 pathogens-10-00823-f003:**
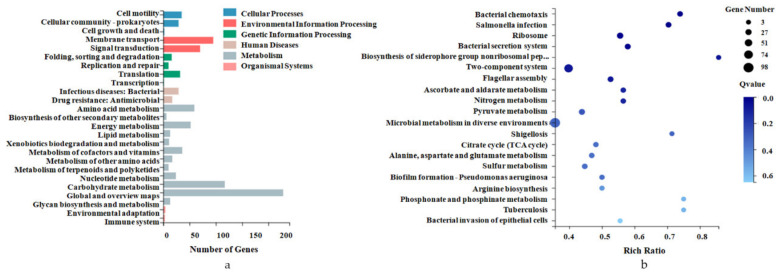
KEGG pathway categories and enrichment analysis. (**a**) The gene-involved KEGG metabolic pathway is divided into seven branches. The X-axis represents the number of genes annotated in a KEGG pathway category, and the Y-axis represents the KEGG pathway category. (**b**) The length of the column at the bottom of the X-axis represents the size of the Q-value (−log10(Q-value)), and the value of the points on the broken line on the top of the X-axis is the number of DEGs annotated on the KEGG pathway.

**Figure 4 pathogens-10-00823-f004:**
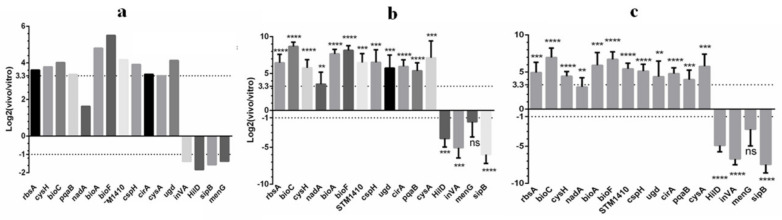
qPCR validation of the RNA-seq data. (**a**) Results are expressed as the target ratio of each gene to log2 (in vivo/in vitro). (**b**) Relative qPCR was used to measure changes in target gene expression in vivo relative to the in vitro expression levels; results are expressed as the target ratio of each gene to log2 (in vivo/in vitro) compared with the reference gene *16S*. (**c**) Relative qPCR was used to measure changes in target gene expression in vivo relative to the in vitro expression levels. Results are expressed as the target ratio of each gene to log2 (in vivo/in vitro) to the reference gene *pdxJ*. Note: * represents a significant difference (*p* < 0.05); ** (*p* < 0.01), *** (*p* < 0.001), and **** (*p* < 0.0001) represent an extremely significant difference; NS means the difference is not significant.

**Table 1 pathogens-10-00823-t001:** The classification of upregulated CKGs.

Gene Description	Number	Gene Symbol
T3SS-associated protein	17	*ssa* (***D***, ***H***, ***I***, *J* [26], *L* [27], *M* [28], *N* [29], ***P***, *S* [30], ***T***, ***U***, and *V* [26,27]), s*sc*A [30,31], s*sc*B [32,33], *sseA* [34,35]), ***yscR*,** *spiA* [36];
Pathogenicity-island-2-secreted effector protein	7	*spiC* [37,38], *sseB* [35], *sseC* [39], *sseD* [40], *sseE* [32,35], *sseF* [35,41], ***STM1410***;
Hypothetical protein	7	***steC***, *SseL* [21,42], ***STM2138***, ***yncJ***, ***ssaK***, p***agO***, ***pagC***
Enzyme	12	***bioF***, *bioB* [43], ***bioA***, ***bioD***, *ugtl* [44,45], *ugd* [46], ***STM1952***, ***cysH***, *sspH2* [47], ***entC***, ***Fe-S***, *pqaB* [48];
Antimicrobial resistance protein	3	*virK* [49], *mig-14* [50], *pagP* [44];
Transporter	5	*mgtC* [51], *mgtB* [52], ***rbsA***, ***cysA***, ***fruB***;
Receptor protein	3	*fepA* [53], *cirA* [54], *iroN* [53]
Cold shock protein	1	***cspH;***
Effector protein	3	***sifB***, *pipB2* [55,56], *pipB* [57];

Note: Attached to the literature are the existing studies that have proven the virulence factors. Bold front indicates CKGs.

## Data Availability

All data generated or analyzed during this study are included in this published article and its Appendix A. The datasets are also available from the corresponding author on reasonable request.

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
