# Peer review of "Analysis of In Vivo Transcriptome of Intracellular Bacterial Pathogen Salmonella enterica serovar Typhmurium Isolated from Mouse Spleen"

_pathogens, 2021, doi:10.3390/pathogens10070823_

Round 1
Reviewer 1 Report
The Authors performed a study that aimed to decipher the genetic elements of S. Typhimurium that are differentially expressed within the murine model of Typhoid fever. BALB/c mice and an IP route of administration were used to identify genes expressed within infected spleens. RNA-Seq and qRT-PCR were used to identify and verify differentially expressed genes, respectively. The presented work is of interest to the field of bacterial pathogenesis, but there are gaps/contradictions within the information. In addition, the work is largely observation-based and not mechanism-based. Addressing the concerns listed below may improve the communication of the work.
- The Introduction is missing relevant information An additional paragraph that addresses the importance of the two most studied SPIs (SPI1 and SPI2) would improve the context for the presented data. In addition, providing information about the BALB/c IP model of Typhoid fever would communicate the reasoning for the in vivo approach.
- Considering the amount of data, the Results section is brief. It would be beneficial if the Authors could provide a section within the Results section that provides details to the information that is highlighted. For instance, pathway enrichment indicated that flagellar assembly was altered across the conditions; however, there is a large set of genes and regulators that control this pathway and no information about these genes are provided.
- It would be beneficial if the Authors could specify the T3SS islands that are differentially regulated (eg., SPI1 vs SPI2, etc).
- Are there CFU/g data associated with the RNA-Seq and qRT-PCR data? This information may be helpful in interpreting data since the infection and likely gene expression can be spatial and temporal.
- Related to #4 above, is there a rationale for harvesting spleens at 2 dpi (line 279)? Can the Authors provide reasoning behind this time point?
- Lines 271-295, can the Authors provide a figure outlining this workflow? This would be helpful in understanding the replicates/pooling strategy used. For instance, line 280 lists 10 BALB/c mice spleens were used, but 3 in vivo samples are presented within the Results section.
- Line 274, for the CFU of the IP inoculum, can the Authors provide the range? As it is listed now, it appears that there was only 1 single inoculum that had a single measurement at 104 CFU.
- Note that there are no control mice spleens included within this study. Is there a justification for not including this control?
- Table S3, the ratio of gene expression is not consistent with the manuscript, in vitro/in vivo instead of in vivo/in vitro. Please use the same comparison for both data sets.
- Could the Authors provide data from the mutant generated with Table S2 primers?
- Discussion section, it would be informative if the Authors can mention specific transcription factors that appear active/inactive based on the gene expression studies. For instance, PhoP/Q, SlyA, Fur appear to be involved in the differentially expressed data.
- Is there information on the transcriptional control of the bio genes? This information would benefit the Discussion section (lines 209 – 219).
- There are no mechanism-based studies shown here. For instance, differential expression of genes may or may not indicate a contribution to infection. Additional in vivo studies with mutant strains in the genes of interest would support the approach used for identification of regulated genes.
Author Response
Thank you for your comments concerning our manuscript entitled “Analysis of In Vivo Transcriptome of Salmonella Typhmurium During Infection of Mice”(ID: pathogens-1273672). Those comments are all valuable and very helpful for revising and improving our paper, as well as the important guiding significance to our researches. We have studied comments carefully and have made correction which we hope meet with approval. Revised portion are marked (No Markup shows how the incorporated changes will look) in the paper. Revised portion are also marked in yellow in the paper. By the way, we can't find “numbered for end” style in the endnote. We temporarily use numbered instead.

Reviewer 2 Report
The manuscript by Sun et al described the analysis of "In Vivo Transcriptome of Salmonella Typhmurium During Infection of Mice".
Comments:
1. Abstract: at the post-NGS era, colleagues generally believe the mRNA expression level in RNA-Seq, the authors do not need to mention "the qPCR results match with the RNA-Seq data" in the abstract
2. Table 1 should be moved to supplementary materials
3. What are the rationales of performing only RNA-Seq of Salmonella infected spleen? Can the authors clarify?
4. As the S. Typhimurium were taken from infected mice and there should be mouse RNA contamination, can the authors clarity the reads mapping % to S. Typhimurium genomes? Are there any reads that map both to mouse and S. Typhimurium genomes?
5. Please check typo and errors across the manusciripts.
Author Response

(The authors gave the same response as above.)
